# Strong coupling and high-contrast all-optical modulation in atomic cladding waveguides

Liron Stern[1], Boris Desiatov[1], Noa Mazurski[1] & Uriel Levy[1]

In recent years, there has been marked increase in research aimed to introduce alkali vapours into guided-wave configurations. Owing to the significant reduction in device dimensions, the increase in density of states, the interaction with surfaces and primarily the high intensities carried along the structure, a plethora of light–vapour interactions can be studied. Moreover, such platform may exhibit new functionalities such as low-power nonlinear light–matter interactions. One immense challenge is to study the effects of quantum coherence and shifts in nanoscale waveguides, characterized by ultra-small mode areas and fast dynamics. Here, we construct a highly compact 17 mm long serpentine silicon-nitride atomic vapour cladding waveguide. Fascinating and important phenomena such as van-der-Waals shifts, dynamical stark shifts and coherent effects such as strong coupling (in the form of Autler–Townes splitting) are observed. Some of these effects may play an important role in applications such as all-optical switching, frequency referencing and magnetometry.

[1] Department of Applied Physics, The Benin School of Engineering and Computer Science, The Center for Nanoscience and Nanotechnology, The Hebrew University of Jerusalem, Jerusalem 91904, Israel. Correspondence and requests for materials should be addressed to U.L. (email: ulevy@cc.huji.ac.il).

The emergence of the field of guided light–vapour interactions, has led to numerous demonstrations of efficient linear and nonlinear light–vapour interactions[1–12]. Motivated by the general trend for miniaturization and high level integration, this effort strives to confine light and vapour in the same quarters to obtain efficient and low-power light–vapour interactions. Integrating vapour cells with optical fibres and waveguides is natural choice for enhancing light–vapour interactions. This is because guided waves do not exhibit light diffraction as an optical waveguide supports tight mode confinement over a long propagation distance. A few approaches for integrating vapours with guided-wave configurations have been demonstrated over the recent years, including the use of hollow-core photonic crystal fibres[3,4,9,10] (HC-PCF), hollow-core anti reflecting optical wave-guides[8,12] (HC-ARROW), tapered nano-fibres[1,2,11] (TNF), surface-plasmon guided resonances[7] and exposed-core fibres[5]. Such integration of vapour with guided systems has served for the demonstration of a myriad of effects ranging from basic linear spectroscopy[1,6,8,13], to a variety of nonlinear effects such as electromagnetic-induced transparency[14] (EIT), enhanced two-photon absorption[2,10], phase switching[4], all-optical modulation[7,15] and slow light[12].

Recently we have demonstrated the atomic cladding waveguide (ACWG), consisting of a silicon-nitride (SiN) core surrounded by a cladding of Rb vapour, which is introduced by integrating an atomic vapour cell above the optical chip[6]. In this demonstration, the ACWG had an interaction length of 1.5 mm and a mode area of $0.3\lambda^2$. With this configuration we demonstrated basic spectroscopy, extremely low saturation power and two-photon transitions. Subsequently, the integration of ACWG based micro-ring resonators has been demonstrated to achieve efficient all-optical modulation[15]. Even more recently, Ritter et al.[16] have implemented such ACWGs in a Mach Zhender interferometer configuration. Such a platform offers several prominent advantages; dimension-wise, an ACWG, has a mode area that is about two orders of magnitude smaller than previously demonstrated on-chip approaches, making it a fine candidate for efficient nonlinear light–vapour interactions. In addition, due to the ability to integrate large variety of existing photonic circuits with the ACWG, this approach can further enhance both linear and nonlinear light–matter interaction via the use of resonators such as micro-ring resonators and photonic crystal resonators, and structures such as slot waveguides. The ACWG SiN platform further enables to couple broadband optical signals, making it applicable (simultaneously) to various alkali vapours such as Cs and Rb and various optical transitions whether the D1 and D2 lines, or higher excited Rydberg states[17].

Here, we introduce the platform of a long serpentine-like atomic cladding waveguide for the purpose of demonstrating two major phenomena: the first, is the ability to obtain coherent effects, using low optical powers for the purpose of all-optical switching. Indeed, we have been able to almost 'switch off' a probe signal, in the presence of a pump beam, having power of only 10 μW. Such switching capability is expected to be relatively fast, as the dynamics of our hot vapour apparatus are in the nano-second regime. The second, is the ability to quantify two important mechanisms for shifts, namely the Van-der-Waals (VDW) shift and the light-shift. Understanding such effects is crucially important for the construction of on-chip frequency references using such platforms. Specifically, owing to our design, which offers high optical density on-chip with an ultra-small footprint we demonstrate linear absorption approaching 25% at temperature as low as 65 °C. Due to the high electromagnetic energy density propagating in the waveguide, we demonstrate that such a platform retains its previous merits, and saturates at the 100 nW regime. Moreover, when introducing a pump-field resonant with a different excited state, we are able to observe strong coupling in the form of a high-contrast Autler–Townes splitting, which can be exploited for highly efficient all-optical switching, using microwatts of power levels. In addition, at such high intensity levels, we observe light shifts of ∼200 MHz. Compared with previous demonstrations, the current device achieves enhanced performance in terms of optical density and non-linearities. As such, it holds an immense promise as a fundamental building block in a variety of light–vapour experiments. Finally, our device offers exceptionally fast (sub ns) transit times, as a result of the extremely small evanescent decay length of the optical mode away from the waveguide core. Such fast transit times accompanied with fast induced Rabi frequencies allow one to observe coherent effects, and fast switching speeds, which are highly advantageous in applications such as all-optical switching.

## Results

**Design of serpentine atomic cladding waveguides.** A sketch of our atomic cladding waveguide is depicted in Fig. 1a, whereas a schematic cross section is shown in Fig. 1b. The ACWG is 17 mm long, and sufficient waveguide spacing of ∼10 μm is used to avoid coupling of light between adjacent pieces of the same waveguide. The bending radius of the curved sections of the waveguide is ∼50 μm. The purpose of such dense packing is to enable as dense as possible interaction region, enabling a small footprint. Note that here we do not investigate the nature of the interaction of a given atom with different sections of the wave-guide (implementing a Ramsey time interferometer), which may be of high significance. This remains a topic of future study. The waveguide is constructed by standard electron-beam lithography followed by reactive ion etching. Its dimensions were designed to support a relative broad spectral range, both in the visible and in the telecom regime. Thus the waveguide supports the D1 and D2 transitions of Rb as well as non-direct transitions at 1,319 and

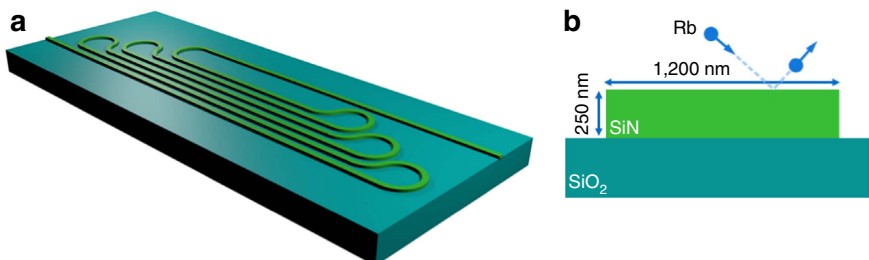

**Figure 1 | Serpentine atomic cladding waveguide. (a)** Sketch of a serpentine atomic cladding waveguide consisting of seven segments of swirled SiN waveguides. **(b)** Cross section sketch of the SiN waveguide, of 250 nm height and 1,200 nm width. Rubidium (Rb) atoms are illustrated as balisitically striking the surface, entering and exiting the evanescent portion of the optical mode.

1,529 nm. Following the definition of the waveguide core we deposit 1 μm thick layer of $SiO_2$ on the entire chip using plasma-enhanced chemical vapour deposition (PECVD). Next we use coarse lithography followed by wet etching to expose the waveguides top cladding. Finally, we integrate the chip with a vapour cell, similarly to the procedure reported in Stern et al.[6] In short, we bond a flat hollow cylinder to the chip with thermal cured vacuum grade epoxy, bake-out and evacuate the bonded cell, introduce [85]Rb into the cell and pinch-off the glass cell from the vacuum system to obtain a portable device.

**Linear spectroscopy and saturation in serpentine ACWGs.**
Next we couple light directly to the ACWG to perform spectroscopic measurements. To do so, we use lensed fibres for coupling light from a fibre-coupled 780 nm laser into the waveguide, and out of the waveguide to the detector. We use the D2 line of rubidium, around 780 nm wavelength. The normalized transmission spectrum of light propagating along the ACWG is shown in Fig. 2a, together with a transmission spectrum of a reference cell. The results were obtained at cell's temperature of about 65 °C, corresponding to atomic density of $3.7 \times 10^{11}$ cm$^{-3}$. The optical power in the waveguide was attenuated to avoid full saturation of the transitions, and is estimated to be about 50 nW. As can be clearly seen, in comparison with the reference cell the absorption lines are broader. As reported[6,16,18], this result is associated with an additional Doppler broadening due to the increased photon momentum along the propagation direction, and also with an increase in the transit time broadening resulted by the limited interaction time of the vapour with the evanescent tail of the optical mode. This broadening manifests as a unified line (consisting of three different unresolved transitions) with a full width half maximum of ~1.17 GHz. The mode content within the waveguide directly affects the Doppler broadening via the effective refractive index. While the exact distribution of modes is unknown and difficult to anticipate, we have slightly simplified the problem by assuming that the electromagnetic energy is equally distributed among the first three modes (the first and second TE modes as well as the first TM mode. Higher modes are assumed to have high bending loss and thus do not contribute significantly). By doing so, the average effective index is ~1.66 (calculated by finite element method simulation), with an uncertainty lower than 5%. As a result, the Doppler-broadened linewidth is estimated to be ~0.9 GHz. Whereas, one dimensional theoretical calculations (Supplementary Fig. 1; Supplementary Note 1) suggest a transit time broadened line of ~100 MHz. As will be evident later, the convolution involved in integrating the susceptibility indeed almost reproduces the measured total linewidth of 1.17 GHz.

Next we compare the spectral position of the dips in respect to the reference cell. Interestingly, we observe a red-shift of approximately 65 MHz. This red-shift may originate from long range VDW interaction between the Rb atoms and the SiN surface. As the average evanescent length of the first three modes is ~90 nm, using an effective interaction length of 45 nm[19], yields a VDW coefficient of 6 KHz μm$^{-3}$, being in the range of VDW coefficients reported previously[20,21]. And yet, when incorporating the VDW frequency shift into the theoretical model (presented in Supplementary Fig. 2 and Supplementary Note 2) using the previously reported value of 1.2 KHz μm$^{-3}$, a shift of 65 MHz is predicted, identical to the measured shift. Such model infers that the effective interaction distance is close to 30 nm. Exploring surface interactions such as VDW shifts is important both from fundamental and applicative aspects[22,23]. For instance, in metrology applications such as optical frequency references, exploring the VDW coefficient for different types of materials, and coatings as well as its environmental (for example, temperature) dependency is highly important. Further work will be devoted to this topic in the future.

Taking into account the above-mentioned effects, we can now fit the obtained spectrum to a model. This model includes transit time broadening, enhanced Doppler broadening, the quenching of atoms on the waveguides wall, the slight saturation of the atoms and the VDW shift and predicts and overall linewidth of 1.15 GHz, very close to the measured linewidth. More details about this model are given in Nienhuis et al.[24] and in Stern et al.[6] In this model, a single fitting parameter is used - the density of atoms (or equivalently, the temperature). The fitting results are displayed in Fig. 2a (see red curve). It is important to note that owing to the use of a relatively long waveguide a high contrast of ~20% is now achievable at a moderate temperature of 65°, which is a significant improvement compared with previously reported results achieved with shorter waveguides. Furthermore, the compact serpentine design allows keeping the footprint of the device as small as possible.

**Coherent light vapour interactions in serpentine ACWGs.**
Next we vary the coupled power in the ACWG and measure its transmission spectrum. Five different measured spectra corresponding to five different power levels are plotted in Fig. 2b. Clearly, it can be seen that varying the power levels changes the transmission spectrum. Specifically, the transition contrasts are diminished for powers of few μWs while for power levels in the nW regime these transitions are recovered. This is a direct consequence of the two-level saturation mechanism. The onset of saturation is estimated to be in the 100 nW regime, consistent with previously reported results[1,2,6]. This low

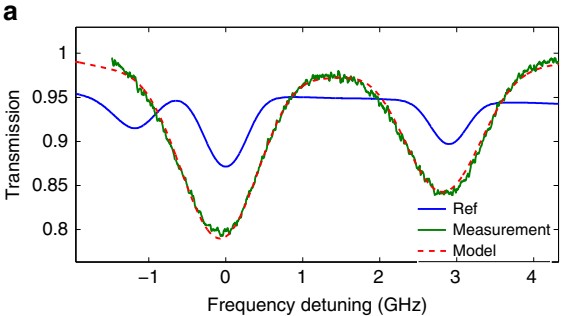

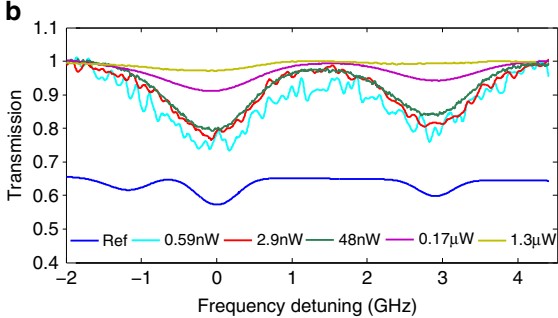

**Figure 2 | Linear spectroscopy and saturation in serpentine ACWGs.** (**a**) Measured [85]Rb transmission spectrum of a 17 mm long ACWG (green), together with the calculated transmission spectrum (red) and the measured transmission spectrum of reference cell containing natural Rb (blue). (**b**) Measured ACWG spectra at different power levels within the waveguide, together with the measured transmission spectrum of reference cell containing natural Rb (blue).

saturation power is due to the high-energy density carried by the guided mode, and serves as an indication for the low-light level needed for other nonlinear process such as two-photon absorption, and EIT. We note that this device has not been optimized for coupling efficiency. Indeed, the device exhibits a loss per facet of ∼20 dB. As a consequence, the low level light traces estimated of having sub nW powers within the waveguide are of pW levels, and comprise of relatively low signal to noise ratio. Coupling efficiency can be further enhanced by adopting known solutions, for example, inverse tapers or grating couplers.

One significant manifestation of such nonlinear process is all-optical switching, that is, the ability to modulate light transmission at a particular wavelength in the presence of signal at another wavelength by a nonlinear mediator. Amongst many different possible schemes for all-optical switching in Rb, we implement here all-optical switching in the D1 and D2 wavelengths. A schematic representation of this process is shown in Fig. 3a, whereas a schematic representation of the optical configuration is presented in Fig. 3b. First, we tune the D2 line of 780 nm to the $F = 3$ to $F' = 2/3/4$ transition, with a pump power in the μW regime within the waveguide. Next, we scan the D1 line (with a relatively unsaturated power level of the order of 100 nW) of 795 nm across the $F = 2/3$ to $F' = 3/2$ manifolds. Both wavelengths co-propagate, and are coupled into the waveguide in the same manner described earlier. The out coupled light, collected with a lensed fibre, is collimated and spectrally filtered using two band-pass filters, operating at the ∼800 nm band, and thus filtering out the 780 nm wavelength. Finally, as before, we collect the filtered signal using a photodetector.

Before elaborating on the results obtained in the waveguide, we briefly describe the results obtained simultaneously using a reference cell (Fig. 3c). Obviously, for this cm-size cell operating with free space beams the intensity levels for both probe and pump are much lower with respect to the waveguide. In Figure 3c, we plot the spectrum of the probe beam with (green line) and without (blue line) the presence of the 780 pump beam. Clearly, we can see that pump beam changes the transmission spectrum

drastically. First, we observe peaks within the $F = 3$ to $F' = 2/3$ transition, attributed to combination of V-type EIT, velocity selective optical pumping and saturation[25,26]. Next, we observe an enhancement of the absorption for the $F = 2$ to $F' = 2/3$ transition, in the form of two dips[7]. We attribute this absorption enchantment to optical pumping which is induced by the pump beam on the $F = 3$ to $F' = 2/3/4$ manifold. We note that these peaks and dips are power broadened, and it should be possible to observe nearly natural line widths (∼6 MHz) using this scheme by reducing the pump power, albeit at the expanse of a lower contrast.

Following this discussion, we can now turn back to describe the atomic cladding waveguide pump-probe spectra. First, we plot in Fig. 3d the transmission spectrum of the probe beam with (green line) and without (blue line) the presence of the 780 pump beam. Here, we use a pump beam with power of ∼10 μW within the waveguide, and an unsaturated probe beam. As can be seen the pump beam strongly modulates the probe spectrum. When the probe is tuned to the $F = 3$ to $F' = 2/3$ line (around zero detuning), we observe a drastic increase in transmission, whereas when the probe is tuned to the $F = 2$ to $F' = 2/3$ line (around 3 GHz detuning), we do not observe any significant change in transmission, and yet observe a significant frequency shift of approximately 200 MHz. We note that for this pump power we are able to almost totally 'shut-down' the absorption of the probe, accompanied with an appearance of a transparency window within the absorption dip. We shall claim that this is a direct consequence of atomic coherence induced by the pump beam. In the case of the $F = 3$ to $F' = 2/3$ transition, the pump depletes the occupancy of $F = 3$ level, and as a result there is less absorption and increased transmission. In contrast, for the $F = 2$ to $F' = 2/3$ transition the atoms, which are pumped from the $F = 3$ level to the $F'' = 2/3$ manifold can spontaneously decay to the $F = 2$ level, and thus increase the population of level $F = 2$ and increase absorption. The magnitude of this phenomena is relatively small due to: (a) The lifetime of 27 ns of Rb is larger than the average time the atom spends in the evanescent region, (b) the branching

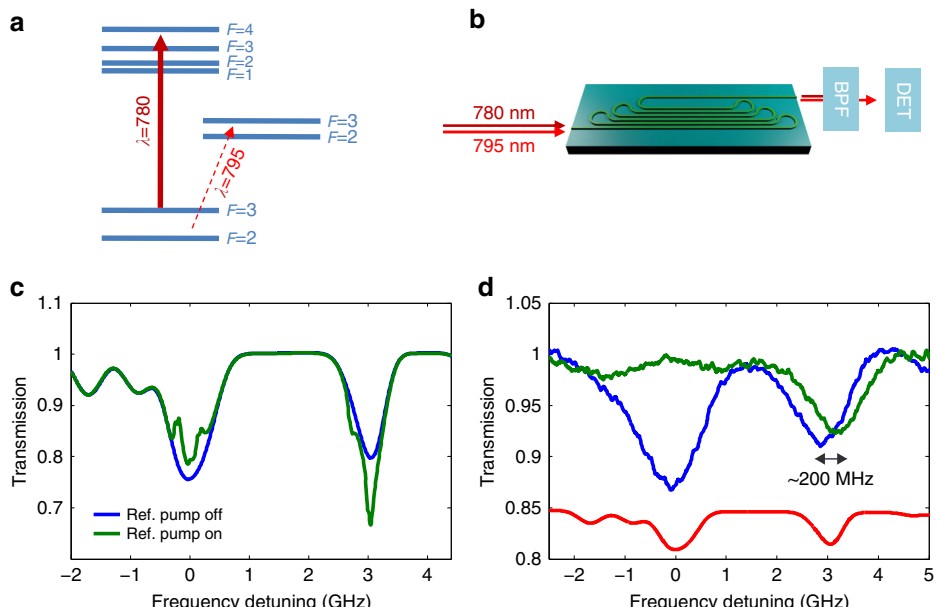

**Figure 3 | nonlinear coherent light–vapour interactions in serpentine ACWGs.** (**a**) [85]Rb relevant transitions of the D1 and D2 manifolds (**b**) Sketch of the optical configuration, illustrating the co-propagating 780 and 795 nm light beams coupled into a serpentine ACWG. (**c**) Measured transmission D1 spectrum (795 nm), propagating through a natural Rb cell reference with and without the presence of a pump beam at 780 nm. (**d**) Measured D1 transmission spectrum, of light propagating through the serpentine ACWG (filled with [85]Rb) with and without the presence of a pump beam at 780 nm. The red line represents a D1 lines reference spectrum.

ratio of decays routs which is favourable towards the $F = 3$ level. The frequency shift, that is observed in the $F = 2$ to $F' = 2/3$ transition is attributed to the dynamical (AC) stark shift, that the pump beam is applying to the $F = 2$ ground state, detuned $\sim 3$ GHz from the pump frequency.

Next, the results obtained for 5 different power levels ranging from 0 to $\sim 13$ μW are presented in Fig. 4a:

As can be observed from Fig. 4a, the contrast of the absorption signal decreases as we increase the power launched in the waveguide, and for the highest power level (blue line, in Fig. 4a), we are able to almost 'turn off' the signal. In the absence of optical pumping and coherent effects the maximal achievable population transfer of the pump can be 50%, and thus the almost 100% absorption reduction seems surprising, as it indicates that either optical pumping or coherent effects (or both) must be present. Yet, optical pumping is of negligible magnitude in our system, mostly due to the short transit time. Indeed, we do not observe any significant optical pumping when the probe beam is tuned from the $F = 2$ to $F' = 2/3$ levels. This is in contrast to the optical pumping observed in the reference cell manifested as enhanced absorption (Fig. 4a black line). Thus we claim, and validate numerically, that the significant reduction in the absorption is attributed to a coherent effect. This reduction in contrast is also accompanied with an appearance of broad transparency window. These finding are validated by calculating (Supplementary Note 3; Supplementary Figs 3–5) the susceptibility of the vapour in the case of a three level system and in the frame work of the spatial dependent Bloch equations in the presence of evanescent electromagnetic fields. Indeed, when one discerns numerically between the coherent (represented by the appropriate off diagonal matrix element) and non-coherent process it turns out that our results can only be explained by the presence of such coherent effects. As our system is governed by fast dynamics, due to the short transit time of the atom in the evanescent region, the observation of such coherent process requires strong intensities, corresponding to Rabi frequencies comparable to the transit time frequency ($\sim$ GHz). Indeed, owing to the highly confined waveguide mode we can achieve Rabi frequencies as high as $\sim 1$ GHz along the entire 17 mm of propagation, with moderate optical power of $\sim 10$ μW within the waveguide.

As mentioned before, the enhanced transparency window is accompanied by a broad peak. This peak is $\sim 1$ GHz wide, and is attributed to a strong coupling effect, that is, the Autler–Townes splitting. Indeed, such an effect is observed for cases of very high Rabi frequency $\Omega_r > \Gamma_d$ where $\Gamma_d$ is the Doppler width. We have observed such peaks very consistently, with varying magnitudes, as a function of the specific pump power within the waveguide.

We note that it may be possible to achieve narrower peaks, in the EIT regime, as has been demonstrated in HC-PCFs[27], simply by reducing the power.

Finally, to quantify the above-mentioned light-shift, we plot in Fig. 4b the $F = 2$ to $F' = 2/3$ frequency shift (obtained by fitting a Gaussian to the measured transmission dip and identifying its centre) as a function of the estimated power which is coupled into the waveguide. The standard expression for the energy shift of an atomic level, induced by an optical field can be calculated using second order perturbation theory. For the case of low intensity relative to the detuning $\Delta$ that is, $\Omega_r^2 / \Delta^2 \ll 1$, a linear[28] dependency between the frequency shift and the optical power is expected. As the detuning in our case is 3 GHz, and the Rabi frequencies (Corresponding to the intensities propagating within the waveguide) vary from the MHz regime to the GHz regime, we expect to find only a slight deviation from the linear regime. By fitting the results to a linear curve we obtain a slope of 13.5 MHz μW$^{-1}$. Using the linear relation between the squared Rabi frequency and the frequency shift we infer a Rabi frequency of 420 MHz per (μW)$^{1/2}$, very close to the estimated theoretical $F = 2$ to $F' = 3$ Rabi frequency of 405 MHz(μW)$^{-1/2}$ (Supplementary Note 4). Moreover, at power of $\sim 13.2$ μW this Rabi frequency of 1.52 GHz is extremely close to the splitting of 1.59 GHz extracted from Fig. 4a. We note that that in case of a Doppler-broadened medium, the splitting is expected to be somewhat larger than the Rabi frequency as evident from Ahmed and Lyyra[29] and our simulations in Supplementary Note 3. As the splitting is governed by a Rabi frequency corresponding to the $F = 3$ to $F'$ manifold, which in average is larger from the $F = 2$ to $F'$ manifold, this agreement between the splitting separation and the inferred light-shift Rabi frequency is justified. Finally, we stress that a huge frequency shift of $\sim 200$ MHz is observed for a moderate power level of only $\sim 13$ μW, and with a relatively large detuning of 3 GHz.

## Discussion

To summarize, we have demonstrated a platform for chip-scale and efficient light–vapour interactions, consisting of compact and long SiN waveguide interacting evanescently with Rb atoms, and used it to study two major phenomena, namely coherent effects (for example, strong coupling) and light and surface dependent frequency shifts.

The coherent effects were observed by the strong modulation of a probe light in response to a pump. With the enhancement of the pump power, we could even reach the regime of strong coupling, manifested as an Autler–Townes splitting.

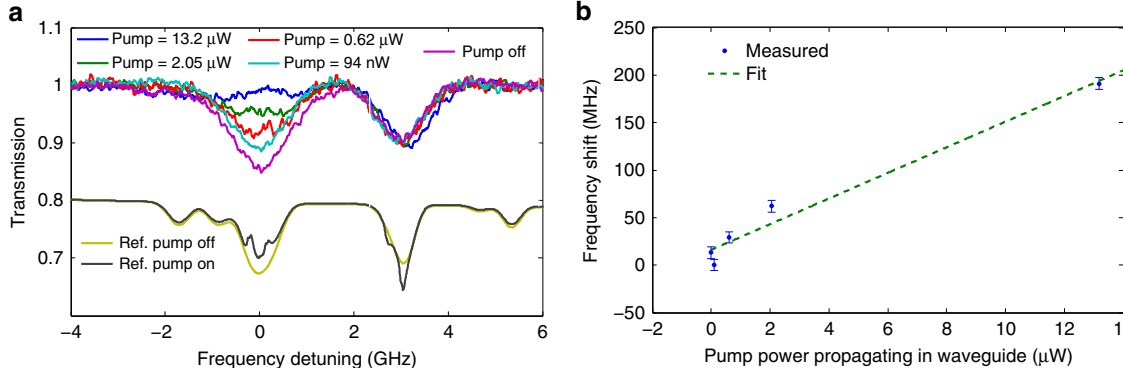

**Figure 4 | Power dependence of transparency window and light-shift. (a)** Measured transmission spectrum of light around the D1 (795 nm) line, propagating through the serpentine ACWG with varying powers in the presence of a pump beam at 780 nm. (**b**) Light-shift: $F = 2$ to $F' = 2/3$ dips frequency shift as function of pump power. Error bars are defined as the fitting error, to the data in **a** of this figure.

This phenomenon is a direct consequence of the light induced coherency in our system. Taking advantage of the tight mode confinement over the entire waveguide length allows us to achieve high Rabi frequencies and all-optical modulation in the µW regime. The speed of such optical switch is in general limited by the lifetime of the $F'' = 2/3/4$ levels (that is, the $5^2P_{3/2}$ manifold). However, this natural lifetime of 27 ns is larger than the time an average atom traverses the evanescent region of the waveguide, being in the ns regime. Thus, effectively, and as has been previously explained by Salit et al.,[30] this would be the time constant for switching corresponding to the hundreds of MHz operation.

Two fascinating mechanisms for frequency shifts were studied. First, we have observed VDW shift of up to 60 MHz. This observation is attributed to the strongly decaying evanescent field outside of the waveguide core. As a result, the field probes atoms which are in close proximity to the SiN surface, experiencing notable long range VDW shifts. We estimate the VDW coefficient to be 6 KHz µm$^{-3}$. Furthermore, by introducing the pump beam we were able to observe light shifts as high as 200 MHz corresponding to a huge slope of $\sim 13.5$ MHz µW$^{-1}$. Studying frequency shifts in such platforms is crucial for the implementation of metrology applications on a chip. Further investigation of the magnitude of such effects in other materials and environmental conditions (for example, temperature) is important for example for understanding the ultimate frequency stability of an optical frequency reference in an ACWG. In particular, it would be interesting to study the VDW shift in micro and nano structured materials, where phonons can be controlled.

Before concluding, we will discuss some of the future prospects in the field. To fully utilize the flexibility of the silicon photonics platform for the purpose of an efficient and integrated light-vapour chip-scale platform, a few technological obstacles remain. First, the device should have a long lifetime, that is, very small rate of vacuum loss and very little outgassing. In addition, to enable scalability and reduced power consumption the volume of the active ACWGs should be minimized. Furthermore, one would like the encapsulation of the chip with a cell to be as compact as possible, for instance by attaching a miniaturized vapour cell to the chip. This could be achieved by anodic bonding a micro machined cell[31] to the surface of the chip. In addition, for many applications, the device should have high optical density even while operating at low atomic density (operating at non elevated temperatures). This will allow This will allow a reduction in the power consumption, increase above-mentioned device longevity and reduce self broadening and shifts[32] accompanied with high density of the vapour. Finally, to fully harness the nonlinear prospects (for example, nonlinear all-optical switching) of ACWGs, long ACWGs are desired. By using such long waveguides, one maintains the confined mode with the accompanied high energy density over long trajectories, and thus accumulate the nonlinear effect. Our platform addresses some of the above-mentioned requirements, as it offers an unprecedented small footprint (effective volume of $5 \times 10^{-4}$ mm$^3$) enables operation with relatively low density of atoms and hence low-power consumption and prolonged lifetime of chip-based integrated vapour systems. In terms of applications, two prominent directions may be envisioned. The first exploits the above-mentioned nonlinear enhancement in the form of low-power all-optical modulator. Numerous systems have been suggested and demonstrated to be able to efficiently modulate light by another light source, where the atomic systems excel in their performance. Specially, we envision that by using the ACWG platform one can achieve a self contained compact hot vapour based system avoiding the complexities accompanied with cold atomic systems. Considering that with HC-PCFs modest switching contrast has been achieved using only 20 photons, and considering both the enabling features coherent interactions impose on all-optical switching as well as the inherent ability to integrate ACWG within resonant elements (such as micro-ring resonators and photonic crystals) the future of few photon all-optical switching in ACWGs seems promising. Another potential application is an ACWG based chip-scale optical frequency standard. Such an optical frequency reference, operating in the Doppler free regime, should provide decent short and long time stabilities. Considering the footprint and power consumption of a device based on the serpentine ACWG, such a device may be an extremely important building block in future photonic circuits. A prominent question remains in respect to the long time stability of such device, where questions this paper addresses such as the magnitude and nature of the VDW and light shifts are of high significance. One direction to improve the short and long time stability, would be to design ACWGs with larger mode areas to reduce both transit time broadening and the VDW shifts. Given the advanced work in the field, we are confident that pushing the boundaries of such technology by having a fully integrated light and vapour system on a chip can accomplished in the foreseeable future.

## Methods

**Fabrication.** To construct the device, we used a silicon substrate having layers consisting of 2.5 µm of thermally grown thick silicon oxide following by a 250-nm thick LPCVD (low-pressure chemical vapour deposited) SiN. After cleaning in Piranha solution (a mixture of sulfuric acid (H$_2$SO$_4$) and hydrogen peroxide (H$_2$O$_2$), in 3:1 ratio). The SiN waveguides were defined using electron-beam lithography (Raith e-line 150) using ZEP-520A as a positive tone electron-beam resist (resist thickness was set to 300 nm). The lithography was followed by reactive ion etching (RIE, Corial) to transfer the pattern into the SiN device layer. After etching, the optical structure was covered with 2 µm thick silicon oxide layer using PECVD. To construct an atomic cell, we carried out an additional lithographic step to define a chamber area where PECVD oxide layer was later on etched down with buffered hydrofluoric acid (BHF) to expose the waveguides structure for interaction with rubidium vapour. Next, the chip was diced to its final dimensions of 1.2 cm. Finally, a Pyrex cylinder was bonded to the chip using thermally cured epoxy (Torrseal). The cell is evacuated and baked out simultaneously. The device was baked out for 24 h, after which, a vacuum level of $10^{-8}$ Torr was reached. At this stage, $^{85}$Rb was launched into the cell, and then, the cell was disconnected and sealed.

**Setup.** Two sources of 780 nm ECDL and 795 nm DFB (Toptica DL-PRO & DL100, respectively) were fibre-coupled to co-propagate a single-mode polarization-maintaining lensed fibre. The polarization direction was set to be aligned with the in-plane (TE) polarization mode of the ACWG. The light was coupled from the lensed fibre into the waveguide in a butt-coupling configuration. Symmetrically, light was coupled out of the waveguide into a second lens fibre using a butt-coupling configuration and the intensity was measured using a photodetector (Newport 2151). For the first part of the experiment, we observe spectroscopy using the 780 nm laser, whereas for the second part we filter out the 780 nm laser (using a two cascaded band-pass filters) and monitor the 795 nm laser. The intensity of the launched light was controlled by inserting ND filters at the output of the laser. In addition both lasers were split before excitation of the waveguide, to measure a reference cell. To achieve sufficient density of atoms, the device was heated using resistor heaters and the temperature was monitored using a thermistor. The thermistor is located a few millimetres away from the top of the cell. To reduce the accumulation of Rb on the surface of the waveguides, we maintained a temperature gradient between the top of the cylinder and the surface of the chip. This way, the Rb droplet is created at the top of the cylinder, where the temperature is lower. The cold spot temperature defines the vapour pressure and consequently, the atomic density.

**Data availability.** The data that support the findings of this study are available from the corresponding author on request.

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

## Acknowledgements

We thank Avinoam Stern, Yefim Barash and Benny Levy from Accubeat for the preparation of rubidium cells, and the use of the vacuum facilities and Tilman Pfau and Robert Löw for fruitful discussions. We would like to acknowledge funding from the ERC grant LIVIN. The waveguides were fabricated at the Center for Nanoscience and Nanotechnology, The Hebrew University of Jerusalem.

## Author contributions

L.S. designed and performed the experiments, analysed the data and wrote the paper; B.D. and N.M. designed and fabricated the SiN waveguides and U.L. supervised the project, designed the experiments and wrote the paper.

## Additional information

**Competing financial interests:** The authors declare no competing financial interests.

