## [Peer Review File · Nature Communications]

Reviewers' comments:

Reviewer #1 (Remarks to the Author):

The authors present results of atomic spectroscopy on an ACWG platform they previously developed. Compared to earlier work, the interaction length has been increased by use of serpentine waveguide. Linear, nonlinear, and coherent spectroscopy are carried out with improved results. This work is inherently incremental, and these types of spectroscopy effects have already been observed in a number of configurations and by several groups, including the authors. No details on the curved waveguide structure are given that would allow for reproduction of the work. The effects of the curvature on waveguide loss and transmission are not discussed. While the results appear sound, they are interesting only to specialists in this community. Therefore, this manuscript is more suitable for a specialized optics journal, see e.g. references 14 or 15.

Reviewer #2 (Remarks to the Author):

Stern et al build on previous work on ACWG to produce two new experimental results

(1) Coherent optical switching in ACWG structures

(2) Quantifying light shift and Van der Waals shifts. The light shift is responsible for (1) above.

Overall, even though experimental technique of ACWG and the details of data collection are not novel to this experiment (see ref 6), the authors demonstrate some new results. The new enabling feature of the current experiment is the increased length of the waveguides, which allows for a longer interaction length.

The observation of Van der Waals interactions close to the surface is not a new physical phenomena (see Ref 20-21), but the authors observe it for the first time in a waveguide structure, which is an important result to document. This is especially true since efforts to use similar configurations for metrology and sensing will require an accounting of all frequency broadening and shifting mechanisms. The observation of coherent light shifts in these structures is more interesting, since the ACWG's, with high mode intensities and short light-atom interaction times, yet long path lengths, seem like ideal candidates to study such effects. I found the results and discussion of the strong-coupling induced coherent Autler-Townes splitting to be the strongest part of the manuscript.

Nonetheless, the authors may wish to be more careful in highlighting the benefits of integrating waveguides with atomic vapors. Efficient nonlinear interaction require both a small field and a large atomic density in the mode. The ACWG certainly provides for a small mode diameter, but suffers from a lack of atomic density in the field. One of the reasons that the optical transitions saturate at low powers is because there are very few atoms in the field.

A few comments/questions about the specific results obtained:

Relating to the Doppler broadening measurements: I calculate a FWHM Doppler broadened linewidth of $\Delta f = n_{\text{eff}} \sqrt{8kT \ln 2 / (mc^2)} = 909 \text{ MHz}$ at 65 C. The manuscript estimates 1.05 MHz of Doppler broadening. Is there a reason for the extra 100 MHz? Perhaps more of the light is in the more highly confined mode with a larger effective refractive index or I have failed to include relevant effect. Perhaps the authors could clarify their calculation in the supplementary information so that others can reproduce it. The manuscript also estimates approximately 250 MHz of transit time broadening. If my above simple calculation is correct, perhaps the transit time broadening might be larger than expected? I get roughly 260 MHz using an rms speed of $v_{\text{rms}} = \sqrt{3kT/m} = 314 \text{ m/s}$ and the

waveguide width of 1.2 μm , which I imagine is how the authors calculated their number. However, the transit time broadening could be increased due to velocity classes traveling in the vertical direction not traversing the waveguides entire width.

Regarding the Van der Waals shift observation: The present manuscript measures a frequency shift in the absorption profile and ascribes it to Van der Waals shift with a VDW coefficient of 6 $\text{kHz}/\mu\text{m}^3$. However, in a very similar configuration (Ref 20) a VDW coefficient next to a sapphire surface is reported to be 1.2 $\text{kHz}/\mu\text{m}^3$ for the Rb $5S_{1/2} \rightarrow 5P_{3/2}$ and 1.9 $\text{kHz}/\mu\text{m}^3$ and for cesium atoms ($5S_{1/2} \rightarrow 6P_{1/2}$) both which match theoretical models fairly well. Ref 21 reports a higher number, but this is for CS atoms near a YAG surface using the CS $6S_{3/2} \rightarrow 6D_{5/2}$ transition, which is known to be stronger owing to the high-lying 6D level. I would suggest that the authors' statement that their measured VDW shift is "in the range of VDW shifts reported previously" may need more substantiation. It would be more convincing, for example, if the authors could account for the increased VDW coefficient prediction (maybe due to the SiN surface? Or lower temperature?)

A couple of other miscellaneous comments

"Ultrafast" is usually thought of as much faster than ns within the optics community. I would suggest using an alternate adjective to describe the observed switching speeds.

What is coupling into/out of lensed fiber? What is waveguide loss? These might be important numbers for future experimental efforts building on this work.

The authors assume that light is equally coupled among first three optical waveguide modes-which ones are these and how is this determination made?

Conclusion:

Overall, I find the most noteworthy result of the manuscript to be the observation of the coherent Autler-Townes interaction. The authors present a reasonable theoretical model that supports their interpretation of the data. I have a slight concern that the result may not be of a general enough interest for publication in Nature Communications, since the results are somewhat specialized. Nonetheless, the quality of the work is high, the references seem appropriately cited, and the conclusions are interesting to me as a specialist in the field.

Reviewer #3 (Remarks to the Author):

The study of tightly confined modes of optical waveguides interacting with atomic vapors is a rapidly growing area of interest for both practical applications and fundamental research. So far, this field includes the use of hollow-core photonic crystal fibers, optical nanofibers, and integrated "on chip" photonic waveguides. Levy's group pioneered the latter. In their earlier work with these ACWG's they used a short (2mm), straight waveguide and demonstrated the technology needed to bond a tiny rubidium vapor cell on top of the chip. They used that system to demonstrate basic rubidium spectroscopy and low power saturation. Here, they extend that work in two significant ways: (1) they use a much longer (17 mm) "serpentine" Silicon Nitride waveguide, and (2) they demonstrate more complex (coherent) interactions between the field and the rubidium vapor. The highlights of these primary advances are (1) The longer waveguide gives them a much larger optical depth; this is significant. (2) the complex interactions include pre-Autler-Townes splitting as a mechanism for low-power (10 μW) all-optical modulation.

In short, I find these advances to be significant and publishable. The coherent nature of the all-optical modulation (a major claim) is justified by calculations that appear to be valid. Additional high-impact features include the discussion of Van der Waals interactions (this is a nascent, but crucial, feature of the entire field).

The paper is well written and the data is convincing. Independent researchers could indeed reproduce the results, given the level of detail provided. The results will be of interest to others in the community. I feel that this paper could be published in Nature Communications after the following points are addressed:

1. In the abstract, the description the "mode volume of $5 \cdot 10^{-13} \text{ m}^3$...over a length of 17 mm" is confusing. This seems to suggest that you want small mode volumes. But in the main text, it seems that you want long lengths, which mean larger "volumes". It would seem to be more relevant to describe it as "the small mode area (of $\ll \lambda^2$) over the long length of 17 mm". It's really the small mode area that is important, over as long a length as possible?
2. Given that the spacing between serpentine straight sections is 10 microns ($d \gg$ evanescent mode; yet, presumably, subject to the same uniform rubidium density), it seems that the same effects could be accomplished in a straight waveguide of length 17 mm. The "compact footprint" argument is used to justify the serpentine geometry, and this seems valid. However, to the broad audience, I fear that the paper seems to imply that the serpentine geometry is necessary. There doesn't seem to be any claim that the serpentine geometry is used to allow the same atom to interact with different regions of the waveguide mode (as it transits over the device-like a Ramsey type effect), which could lead to interesting physics. So it seems important to emphasize that the presented results could also be obtained with a 17 mm straight waveguide.
3. The low power traces (in Fig. 2, 3, 4) are very noisy. This is probably just a SNR issue in the measurement method (the higher power tracers don't look as noisy), and not a fundamental issue. If that's the case, it would help clarify things to mention that it's just SNR.
4. Concluding the VDW paragraph with "further work will be devoted to this topic..." seems inadequate. Could there at least be some additional references included in the previous two sentences on the importance of VDW for fundamental science and for metrology applications?
5. On page 8, the claim of "Taking advantage of the tight mode confinement.....allows us to achieve high Rabi frequencies and all optical switching..." seems misleading. The data shows modulation, not switching.
6. Also on page 8, the statement "Studying frequency shifts in such platforms is detrimental for the implementation of metrology applications on a chip" is very confusing.
7. Ref. 8 and 12 are the same.

Reviewers' comments:

Reviewer #1 (Remarks to the Author):

The authors present results of atomic spectroscopy on an ACWG platform they previously developed. Compared to earlier work, the interaction length has been increased by use of serpentine waveguide. Linear, nonlinear, and coherent spectroscopy are carried out with improved results. This work is inherently incremental, and these types of spectroscopy effects have already been observed in a number of configurations and by several groups, including the authors. No details on the curved waveguide structure are given that would allow for reproduction of the work. The effects of the curvature on waveguide loss and transmission are not discussed. While the results appear sound, they are interesting only to specialists in this community. Therefore, this manuscript is more suitable for a specialized optics journal, see e.g. references 14 or 15.

Our comment:

In response to the comment by the reviewer, we now provide the details regarding the curved waveguide structure (bending radius of $50\mu\text{m}$ and separation between adjacent waveguides of $10\mu\text{m}$). As for the effect of the curvature on waveguide loss – it is negligible because we are using waveguide with high index contrast. In fact, this is another advantage of our system compare with other guided wave integrated atomic systems. We now explicitly mention this in the text.

As for the general comment regarding the significance of our work, this is of course a matter of personal taste. We absolutely believe that this paper is far from being incremental. It provides few very important contributions and it was also recognized by the other two reviewers. We would like to re-address again the novelty and the significance of the paper:

1 - Significance – We can now can answer for the first time the open question "is it possible to see coherent effects (e.g. strong coupling) in chip scale systems". Taking advantage of the above, we actually demonstrated highly efficient all optical switching. Such a high efficiency is a direct consequence of the unique geometry and configuration of our system. Furthermore, there is a wealth of physics which leads to such a striking result. In that sense, the atomic cladding wave guide provides a unique environment which enables one to observe such effects. The observed spectroscopic features are a direct consequence of the fast dynamics of atoms within the 100nm evanescent wave (faster than relaxation times) and fast dynamics induced by light (in terms of Rabi frequencies) as a consequence of tight confinement.

2 - Practicality - our device provides the longest on chip interaction length between light and vapor, more than order of magnitude with respect to our previously reported device. As such, we can now operate the device at room temperature and still get significant results. In fact, around 60 degrees (C) we can now achieve about 25% contrast. It means that the current device is well within the specifications of many atomic systems (frequency standards, clocks, etc.). This is super important from the application point of view. Furthermore, being able to confine the optical mode over

such lengths is a record in chip scale light vapor interactions. As such, nonlinear effects are being pushed to the extreme in terms on the minimal optical power that is needed.

3 - Finally, we have observed here for the first time light shift and VDW shift in chip scale vapor cells. This has an enormous significance, as it defines the opportunities and the limitations of the chip scale approach.

Reviewer #2 (Remarks to the Author):

Stern et al build on previous work on ACWG to produce two new experimental results
(1) Coherent optical switching in ACWG structures
(2) Quantifying light shift and Van der Waals shifts. The light shift is responsible for (1) above.

Overall, even though experimental technique of ACWG and the details of data collection are not novel to this experiment (see ref 6), the authors demonstrate some new results. The new enabling feature of the current experiment is the increased length of the waveguides, which allows for a longer interaction length.

The observation of Van der Waals interactions close to the surface is not a new physical phenomena (see Ref 20-21), but the authors observe it for the first time in a waveguide structure, which is an important result to document. This is especially true since efforts to use similar configurations for metrology and sensing will require an accounting of all frequency broadening and shifting mechanisms. The observation of coherent light shifts in these structures is more interesting, since the ACWG's, with high mode intensities and short light-atom interaction times, yet long path lengths, seem like ideal candidates to study such effects. I found the results and discussion of the strong-coupling induced coherent Autler-Townes splitting to be the strongest part of the manuscript.

Nonetheless, the authors may wish to be more careful in highlighting the benefits of integrating waveguides with atomic vapors. Efficient nonlinear interaction require both a small field and a large atomic density in the mode. The ACWG certainly provides for a small mode diameter, but suffers from a lack of atomic density in the field. One of the reasons that the optical transitions saturate at low powers is because there are very few atoms in the field.

Our comment: We agree. In response we now mention this issue and discuss it (see discussion section).

A few comments/questions about the specific results obtained:

Relating to the Doppler broadening measurements: I calculate a FWHM Doppler broadened linewidth of $\Delta f = n_{\text{eff}} \sqrt{(8kT \ln 2 / (mc^2))} = 909 \text{ MHz}$ at 65 C. The manuscript estimates 1.05 MHz of Doppler broadening. Is there a reason for the

extra 100 MHz? Perhaps more of the light is in the more highly confined mode with a larger effective refractive index or I have failed to include relevant effect. Perhaps the authors could clarify their calculation in the supplementary information so that others can reproduce it. The manuscript also estimates approximately 250 MHz of transit time broadening. If my above simple calculation is correct, perhaps the transit time broadening might be larger than expected? I get roughly 260 MHz using an rms speed of $v_{\text{rms}} = \sqrt{3kT/m} = 314$ m/s and the waveguide width of 1.2 μm , which I imagine is how the authors calculated their number. However, the transit time broadening could be increased due to velocity classes traveling in the vertical direction not traversing the waveguides entire width.

Our response: Thank you very much for your valuable comment. We have now added a supplementary section addressing the calculations of transit time broadening and Doppler broadening. We also update the numbers in the manuscript accordingly, as we have erroneously estimated a 1.3 GHz FWHM (From Fig. 2a, corresponding to all three transitions together) instead of 1.17 GHz. The theoretical line shapes yield a FWHM of 1.15 GHz. For a single transition, the contribution of TTB is ~ 100 MHz and for the Doppler is ~ 900 MHz (Interestingly they do not sum together due to the nature of the double velocity convolution integration).

Regarding the Van der Waals shift observation: The present manuscript measures a frequency shift in the absorption profile and ascribes it to Van der Waals shift with a VDW coefficient of 6 KHz/ μm^3 . However, in a very similar configuration (Ref 20) a VDW coefficient next to a sapphire surface is reported to be 1.2 KHz/ μm^3 for the Rb $5S_{1/2} \rightarrow 5P_{3/2}$ and 1.9 KHz/ μm^3 and for cesium atoms ($5S_{1/2} \rightarrow 6P_{1/2}$) both which match theoretical models fairly well. Ref 21 reports a higher number, but this is for CS atoms near a YAG surface using the CS $6S_{3/2} \rightarrow 6D_{5/2}$ transition, which is known to be stronger owing to the high-lying 6D level. I would suggest that the authors' statement that their measured VDW shift is "in the range of VDW shifts reported previously" may need more substantiation. It would be more convincing, for example, if the authors could account for the increased VDW coefficient prediction (maybe due to the SiN surface? Or lower temperature?)

Our response: Thank you for this very important comment! Indeed our VDW coefficient is higher than those reported elsewhere. This most likely originates the estimation of the effective distance where the atoms contribute mostly to the signal. We have used an estimation of $L/2$, adopted from calculations in thin cells. And yet, the case of an evanescent field interaction is obviously different than that of a thin cell. As the VDW effect scales cubically, the difference between 30nm evanescent length and 45nm yields a significant difference in the VDW coefficient. Indeed, we now have conducted an additional calculation, were we exploit the fact that in order to calculate the effective susceptibility of the atoms one is in need to integrate the atomic response as a function of distance from the surface. By adding the VDW shift to the atomic transitions energy, we can now obtain the shifted lineshape. When doing so, and assuming a VDW coefficient of 1.2 KHz μm^3 (as is for we the Rb $5S_{1/2} \rightarrow 5P_{3/2}$ with sapphire) we get a line shifted by 65 MHz as in our measurement. We now add this calculation to the supplementary, and have revised the text accordingly.

We would like to stress that although this agreement is quite appealing, more effort should be invested in this important subject, were one should systematically investigate the VDW shift, and changes various parameters such as temperature and evanescent length.

A couple of other miscellaneous comments

"Ultrafast" is usually thought of as much faster than ns within the optics community. I would suggest using an alternate adjective to describe the observed switching speeds.

Our response: We agree, and have changed the adjective.

What is coupling into/out of lensed fiber? What is waveguide loss? These might be important numbers for future experimental efforts building on this work.

Our response: We have added these numbers to the manuscript.

The authors assume that light is equally coupled among first three optical waveguide modes-which ones are these and how is this determination made?

Our response: The first three modes are the TE first mode the TE second mode and the TM first mode. These modes all have effective index in the range of 1.73-1.61, higher modes also exist and yet as they are less confined they are expected to have higher bending loss. It is highly difficult to anticipate what is the exact modal content within the waveguide, and thus we assume an equal distribution for simplicity. We add a short discussion regarding this comment.

Conclusion:

Overall, I find the most noteworthy result of the manuscript to be the observation of the coherent Autler-Townes interaction. The authors present a reasonable theoretical model that supports their interpretation of the data. I have a slight concern that the result may not be of a general enough interest for publication in Nature Communications, since the results are somewhat specialized. Nonetheless, the quality of the work is high, the references seem appropriately cited, and the conclusions are interesting to me as a specialist in the field.

Reviewer #3 (Remarks to the Author):

The study of tightly confined modes of optical waveguides interacting with atomic vapors is a rapidly growing area of interest for both practical applications and fundamental research. So far, this field includes the use of hollow-core photonic crystal fibers, optical nanofibers, and integrated "on chip" photonic waveguides. Levy's group pioneered the latter. In their earlier work with these ACWG's they used a short (2mm), straight waveguide and demonstrated the technology needed to bond

a tiny rubidium vapor cell on top of the chip. They used that system to demonstrate basic rubidium spectroscopy and low power saturation. Here, they extend that work in two significant ways: (1) they use a much longer (17 mm) "serpentine" Silicon Nitride waveguide, and (2) they demonstrate more complex (coherent) interactions between the field and the rubidium vapor. The highlights of these primary advances are (1) The longer waveguide gives them a much larger optical depth; this is significant.

(2) the complex interactions include pre-Autler-Townes splitting as a mechanism for low-power (10 uW) all-optical modulation.

In short, I find these advances to be significant and publishable. The coherent nature of the all-optical modulation (a major claim) is justified by calculations that appear to be valid. Additional high-impact features include the discussion of Van der Waals interactions (this is a nascent, but crucial, feature of the entire field).

The paper is well written and the data is convincing. Independent researchers could indeed reproduce the results, given the level of detail provided. The results will be of interest to others in the community. I feel that this paper could be published in Nature Communications after the following points are addressed:

1. In the abstract, the description the "mode volume of $5 \cdot 10^{-13} \text{ m}^3$...over a length of 17 mm" is confusing. This seems to suggest that you want small mode volumes. But in the main text, it seems that you want long lengths, which mean larger "volumes". It would seem to be more relevant to describe it as "the small mode area (of $\ll \lambda^2$) over the long length of 17 mm". It's really the small mode area that is important, over as long a length as possible?

Our response: We agree that this is a source of confusion. Indeed there are two seemingly contradictory goals: having a long length, and at the same time maintaining the smallest footprint as possible. And yet, that is one of the main reason we have constructed a serpentine ACWG which waveguides are stacked together as tight as possible. We have further clarified this issue at the main text, and in the abstract.

2. Given that the spacing between serpentine straight sections is 10 microns ($d \gg$ evanescent mode; yet, presumably, subject to the same uniform rubidium density), it seems that the same effects could be accomplished in a straight waveguide of length 17 mm. The "compact footprint" argument is used to justify the serpentine geometry, and this seems valid. However, to the broad audience, I fear that the paper seems to imply that the serpentine geometry is necessary. There doesn't seem to be any claim that the serpentine geometry is used to allow the same atom to interact with different regions of the waveguide mode (as it transits over the device-like a Ramsey type effect), which could lead to interesting physics. So it seems important to emphasize that the presented results could also be obtained with a 17 mm straight waveguide.

Our response: We agree, and have added a small paragraph to reflect this important comment.

3. The low power traces (in Fig. 2, 3, 4) are very noisy. This is probably just a SNR issue in the measurement method (the higher power tracers don't look as noisy), and not a fundamental issue. If that's the case, it would help clarify things to mention that its just SNR.

Our response: We agree and have added a comment which reflects this.

4. Concluding the VDW paragraph with "further work will be devoted to this topic..." seems inadequate. Could there at least be some additional references included in the previous two sentences on the importance of VDW for fundamental science and for metrology applications?

Our response: We have added references that reflect the importance of VDW for both topics.

5. On page 8, the claim of "Taking advantage of the tight mode confinement.....allows us to achieve high Rabi frequencies and all optical switching..." seems misleading. The data shows modulation, not switching.

Our response: We agree and have changed this.

6. Also on page 8, the statement "Studying frequency shifts in such platforms is detrimental for the implementation of metrology applications on a chip" is very confusing.

Our response: We agree and have fixed this.

7. Ref. 8 and 12 are the same.

Our response: We agree and have fixed this.

REVIEWERS' COMMENTS:

Reviewer #2 (Remarks to the Author):

The authors have made valuable additions/changes to the manuscript as a result of the referees comments. They have addressed all of my concerns to my satisfaction, though still have a concern that the result may not be of a general enough interest for publication in Nature Communications, since the results are somewhat specialized.

Reviewer #3 (Remarks to the Author):

I have now read the other reviewers reports, the authors' response to those reports (and my own report) and the revised manuscript.

In the revised manuscript the authors have addressed all of my concerns adequately. I feel that they have also adequately addressed the concerns of the other two reviewers. I stand by my original conclusion that the paper can be published in Nat. Commun.

REVIEWERS' COMMENTS:

Reviewer #2 (Remarks to the Author):

The authors have made valuable additions/changes to the manuscript as a result of the referees comments. They have addressed all of my concerns to my satisfaction, though still have a concern that the result may not be of a general enough interest for publication in Nature Communications, since the results are somewhat specialized.

We thank the reviewer for his comments. We strongly believe that the results are of great interest to the broad scientific community. Particularly, the demonstrated results are expected to have implications in various fields involving light-matter-interactions, and atom-like systems.

Reviewer #3 (Remarks to the Author):

I have now read the other reviewers reports, the authors' response to those reports (and my own report) and the revised manuscript.

In the revised manuscript the authors have addressed all of my concerns adequately. I feel that they have also adequately addressed the concerns of the other two reviewers. I stand by my original conclusion that the paper can be published in Nat. Commun.

We thank the reviewer for his comments.